# Cycling Tourism and Revitalization in the Sicilian Hinterland: A Case Study in the Taormina–Etna District

**Gianni Petino** [1,*], **Giuseppe Reina** [1] **and Donatella Privitera** [2]

1  Department of Political and Social Sciences, University of Catania, 95131 Catania, Italy; giuseppereina72@gmail.com
2  Department of Educational Sciences, University of Catania, 95124 Catania, Italy; donatella.privitera@unict.it
*  Correspondence: gianni.petino@unict.it

**Abstract:** This study aims to present a strategy for the revitalization of the Sicilian "internal areas", recognizing a directional tool, together with the integration of self-centered actions of slow tourism. The design was specifically located in the Taormina–Etna tourist district (an area of northeastern Sicily that includes 60 municipalities) which, in rethinking the post-pandemic restart, aims at the development of a mobile system of cycling tourism able to interconnect cultural peculiarities, environmental characteristics, and landscape values. This paper also examines key features and interpretations, and develops a strategy based on a slow travel framework as an alternative means of achieving success in the Sicilian hinterland. Starting from the current financial and environmental crisis, therefore, the paper finds explanations and solutions, in which we try to conceive of the economy and ecology as systems that not only open to one another, but mutually determine one another in defining new, self-sustaining local development processes. In order to build a competitive alternative to help less favorable regions, it is necessary to move within the scope of investments by a public system capable of planning resilient strategies based on sustainable principles.

**Keywords:** thematic itineraries; cycle tourism; experiential tourism; strategic planning

## 1. Introduction

The significant reduction in the demand of world tourism following the COVID-19 pandemic is pushing towards a deep economic crisis not only for local communities but also for all of the operators involved in the tourism sector, including the states and their reception policies. In the wake of the pandemic, the desire to travel will not change in the medium term, and will remain relevant; however, the impact of COVID-19 might stimulate the use of sustainable tourist activities, along with measures and instruments for government monitoring of their development. Consequently, the status quo of tourism destinations could be better reflected in certain aspects, and serve as a basis for decision making in politics, as well as for a regular screening of tourism and the environment.

In addition, the main objective remains to ensure that the territories (in particular the internal areas) and, therefore, the human communities are perfectly integrated into the ecosystems that surround them, in an effort to minimize their ecological footprint in line with the Sustainable Development Goals (SDGs). Starting from the current financial and ecological crisis, therefore, this paper aims to find reasons and solutions for a new geography, wherein to conceive the economy and ecology as systems that not only open to one another, but mutually determine one another in the implementation of territorial cohesion strategies.

The starting point for building a competitive alternative is to move within a public system capable of planning sustainable strategies with local stakeholders by seeking new paradigms of self-development that can help to overcome the current crisis in more marginalized areas. This study aims to represent a strategy for the revitalization of

the Sicilian "internal areas", recognizing the actions of slow tourism (referred to as slow) as a directional tool.

In rethinking the restart, there is the need to enhance post-pandemic tourism, inserting a strategic choice with the aim of enhancing the potential of the Taormina–Etna tourist district (an area that includes 60 municipalities)—in particular by means of cycling tourism, as a generator of new ecologically interconnected landscapes.

In order to achieve the intended objective, it is necessary to grasp the strategies to be implemented by public administrators in the context of territorial marketing policies for the composition of the tourist offer (e.g., commercial activities for bicycles, cycling workshops). The article is based on a multimethod approach that includes observation in the field, as well as in-depth studies aimed at evaluating the perceptions of internal and external stakeholders through a qualitative perspective, based on semi-structured interviews with institutional actors, entrepreneurs, and representatives of local communities. This paper is structured in two parts: In the first part, an attempt is made to approach the theoretical background of the development of slow and cycling tourism as opportunities for the development of entrepreneurial activities (and as a useful response to the crisis faced by sectors such as tourism that have been strongly affected by COVID-19). In the second part, an analysis of the area of the municipalities of the Taormina–Etna District (DTE) is carried out, identifying and mapping the municipalities by their prevalent tourist categories. Finally, the study proposes a development strategy for the territories based on the basic idea of cycling tourism.

## 2. Materials and Methods

### 2.1. Values and Prerogatives of Slow Tourism

Slow movement is a lifestyle adopted in different areas, from the gastronomic, to the cultural and urban, and can also characterize tourism. According Yurtseven and Kaya [1], slow tourism has two main purposes: for tourists to take their time, and to become attached to the place visited. Emphasis is placed on the element of time in order to deepen the knowledge of a destination, where the dimension of consciousness—understood as knowledge of the self and others—takes on importance in a renewed relationship between the local community, oneself, and the culture of the place(s) visited [2,3]. In the existing literature, characteristics are known and established as being attributable to tourism where the slow component have different dimensions, by identifying the following: contamination by other types of tourism, which finds authenticity in the visiting of the places and the enjoyment of cultural heritage [4]; sustainability with respect to the environment, by reducing speed, thus preferring mobility by means of transport with a minimal environmental impact (walking, cycling, etc.); time to enjoy the beauty of nature and the landscape, but also contact with local people, becoming a traveler and not just a spectator [5]. Slowness has the potential to change perceptions and attitudes towards places and nature. Finally, the appreciation of tourism is a relevant component of sustainable tourism [6–8], but even more so of ecotourism [9] and bicycle tourism.

Slow tourism (and everything to which it is connected) represents a tool for the sustainable development of even remote and marginal destinations—including slow territories—i.e., areas with low population density, with a significant rural context, where there are important heritages of tangible and intangible resources [10]—bringing together tourism operators, institutions, and users, and helping to generate widespread benefits for the territory together with environmental protection. To this end, it seems appropriate to stimulate the dissemination of sustainable practices attentive to environmental, social, cultural, and economic impacts, highlighting the importance and validity of the concept of sustainable tourism in line with the increased attention to the long-term vision of rural areas characterized by opportunities, innovation, modernity, vibrancy, resilience, and equality, as well as sustainable and multifunctional environments [11].

Hinterlands—often rural and marginal areas—are characterized by a unique value from a naturalistic and environmental point of view, as well as in relation to the traditions

and historical heritage of the local communities that reside there. Such territories promote the therapeutic potential of nature, as ascertained by multiple studies [12–14] showing that landscapes offer a therapeutic component representative of holistic connections between nature, the self, and physical, social, and mental wellbeing, hence the need for attention to the purposes of economic and social development of these territories, as they are often characterized by natural, structural, and/or demographic handicaps as a result of their remoteness.

### 2.2. Sustainable Mobility and Cycling Tourism

Scholars have addressed the issue of slow tourism and cycling, in studies involving both a perspective of territorial enhancement that enhances these components, as well as natural, rural, and therapeutic territories, and the wellbeing arising from the exercise of cycling [13]. The bicycle has been present for many years, and is currently back in vogue because it can mitigate the anxieties arising from COVID-19, as a therapeutic tool that encourages mindfulness. In addition, bicycle travel can offer less restrictive feelings than other forms of mobility, and in some places represents a traditional mode of transportation (e.g., rickshaws in Asian countries) preserved as a part of local identity, and as a symbol of the slow past, evoking nostalgic memories [14]. In addition, cycling provides opportunities for leisure and recreational activities in sustainable destinations—very often uncrowded—where tourists discover territories with important tangible and intangible heritages [15].

Cycling tourism can contribute positively to health and wellbeing, even within the span of several days [16], and it also has social and environmental benefits [17] and positive impacts that facilitate contact with nature and local people, as well as access to places off the beaten path or remote territories [18]. The itinerary of a cycling ride is based on immersion in the territory—often rural countryside, but also coastal.

There are many direct and indirect effects related to the spread of travel or excursions by bicycle. Under the landscape profile, the aim is to enhance and redevelop areas through the increase in the wide range of ecosystem services provided by the natural territories involved, which also aim at creating new opportunities and qualified employment, with integrative values compared to the already existing tourist offers. Although studies show benefits of bicycle tourism by greenways, trails, and bike paths, none of this directly quantifies the activity, i.e., there is no guarantee that the use of such infrastructure is associated with an actual increase in cycling activity [19]. Undoubtedly, many low-speed roads appear to be safe, and are functional for cycling activities, even if they are not designated for such (i.e., cycleways), and bicycle tourists prefer areas with calm traffic [20] combined with landscapes immersed in nature, where the connection between nature, landscape, and agro-sylvo-pastoral activities is one of the elements of attractiveness and value of the area explored.

Cycling tourism can take on different modes, and implies activity and/or passivity at the same time as a different leisure culture—long-distance travel alone or in organized groups, short bike tours, hikes or real explorations of urban and rural sites—for both short and long recreational activities, but not for sports, although it also includes visual participation in cycling events [21]. Bicycle tourists use bicycles as their primary means of transportation in recreational activity, although not necessarily exclusively, depending on whether they are "enthusiastic" or "accidental" bicycle tourists—with reference to parameters of time and the purpose of the vacation—or even amateur athletes. When talking about cycling tourism, it is appropriate to distinguish between the different targets to which it is addressed because, according to the degree of specialization in the activity and the type of trip [22], the preferences and needs of the cyclist change, with particular reference to the services required [23,24].

The cycling product is combined with other products for a complete and varied tourist experience: according to Isnart-Legambiente [25] (2020), it is combined with visits to cities and towns, food and wine (100%), culture and wellness (71%), and the sea (57%). It assumes

a certain length and daily stages that, depending on the different cyclists (amateurs who use their bikes for the purpose of the trip and/or specialists who use their bikes to perform physical activity and sports), should not be too long (about 50–60 km), with overnight stays in facilities that have appropriate services (e.g., parking, repair shops, etc.). Usually, the length of the trip for an Italian cyclist lasts from 2 to 3 days, up to a maximum of 5 days for foreign tourists; in recent times this is a predominantly national demand, but before the pandemic it was exactly the opposite, involving a variety of markets, both Eurowestern and non-European [26].

Specifically, in the cycling tourism market in Italy, according to Isnart-Legambiente (2020), indications emerge of a developing phenomenon, growing by about 30% in the past five years since, although niche, it is a sector with exponential potential. As an indication, during 2019, cycling tourism generated ~55 million overnight stays, equal to 6.1% of those recorded overall in Italy, generating a total revenue of EUR 4.6 billion. Tourists who choose to spend their vacations by bike—mainly in couples without children in tow (41% of the total)—opt for the northern regions (Trentino Alto Adige, to which 30% of cycle tourists turn, followed by Lombardy (14%) and Veneto), which offer, among other things, kilometers of quality bike paths and numerous accessible services. The residual southern area (Sardinia, Apulia, Calabria, and Sicily) is nevertheless appreciated, accounting for 10% of bicycle tourists. Bicycle tourists are primarily adults over the age of 51 followed by the 26–50 age group (26%) and those under 26 (7%). In recent times, as a result of the related subsidy, there has been a rediscovery of the use and demand for e-bikes or pedal-assisted bicycles, although only 9% of organized tours have requested them.

In cycling tourism, the implementation of cycle paths for mainly recreational and touristic purposes is becoming a need increasingly felt by many jurisdictions, which are aware of Italy's infrastructure deficit compared to other European countries. The FIAB (Italian Federation of Friends of the Bicycle) has proposed a bicycle network of supra-regional scale, and with a connection to neighboring countries, following the model already successfully implemented in Germany, Austria, and Holland for the implementation of the EuroVelo project. Overall, ~17,000 km of cycle network is planned on 17 routes, 3 of which are already part of the EuroVelo cycle routes, and are divided into "Great Ways" and the "Ways of the Two Seas", which should connect the whole country, including the islands.

In addition to upgrading the bicycle infrastructure, there is a need for the management of intermodality, i.e. the integration between the various public means of transportation by citizens and tourists, which could have an impact on the spread of bicycle use. If we look at bike–train intermodality in Italy, the "Ferrovie dello Stato" website specifies that "on Regional trains expressly indicated in the Official Timetable, it is possible to transport a bicycle (maximum one per traveler and no longer than two meters)". Groups of at least 10 people wishing to transport as many bicycles must make an explicit request to the Regional Directorate (www.trenitalia.com, accessed on 12 July 2021).

However, it should be noted that the presence of bike lanes is not an essential element in structuring a bicycle touring route; the Highway Code (Legislative Decree 285 of 30 April 1992, and subsequent amendments) adds to the definition of bike paths in article 3, section 1, point 39: "the longitudinal part of the road, suitably delimited, reserved for the circulation of bicycles" as the definition of the road type F-bis "bicycle and pedestrian routes" (article 2, section 2, introduced by Decree 285 of 30 April 1992, and article 2, section 2, introduced by Decree Law 27 June 2003, n. 151, converted, with amendments, in the law of 1 August 2003, n. 214), defined by section 3 of the same article as a "local road, urban, suburban or proximate, intended primarily for pedestrian and bicycle traffic and characterized by an intrinsic safety to protect the weak users of the road".

*2.3. The Importance and Enhancement of the Remote Territories of Inner Areas*

It remains difficult to trace back to when these dynamics of economic depletion and marginalization began, and who or what triggered them. However, it appears true that this occurs in a longitudinal and transversal sense, and affects the whole national territory,

even if some southern regions are more refractory than others to the attempts at revitalization that took place during the more recent history of the republic [27]. In an attempt to understand the reasons for this, and to trigger possible revitalization, even the political lexicon is increasingly enriched with the frequent use of the terms "disadvantaged areas" [28] and "inland areas" [29], until the final formalization, in 2012, of the launch of the National Strategy for Internal Areas (NSIA)—a national planning tool that provides for a reinterpretation of the Italian territory according to the presence and usability of the services to "citizenship" [30]. Inner areas have been defined as territories characterized by depopulation and decay, and at some considerable distance from hubs offering essential services, such as education, health, and mobility; nevertheless, they are also recognized as playing an essential role in territorial cohesion as key factors of spatial justice, and central to defining the right to citizenship. A number of groups of peripheral municipalities were selected to test pilot projects of socioeconomic development [30,31]. The demographic statistics outline a picture in which over the last 20–30 years there have been dozens of municipalities whose population has halved, and it is equally alarming that, in the next 10–20 years, entire communities could disappear, causing serious impoverishment in terms of historical cultural heritage and traditions. The depopulation of inner areas is also worrying in terms of the related recessive effects on the local economy of the disappearance of professionally qualified figures and commercial activities in many Sicilian municipalities.

The NSIA, based on a wide set of indicators, has made it possible to classify and map all Italian municipalities (7903, to date) according to the availability of services related to health, education, and mobility, or the travel time necessary for their use, thus categorizing municipalities as "poles" when services are present, as "belt" municipalities when the travel time to reach them will be less than 20 minutes, as "intermediate" municipalities when the time will be between 20 and 40 minutes, as "peripheral" municipalities when the time needed to reach them will be between 40 and 75 minutes and, finally, as "ultra-peripheral" when the time required will be greater than 75 minutes. What certainly emerges, in short, is that territories are populated according to the presence or proximity of the services necessary to lead a dignified life.

The NSIA has put in place, and is further refining, territorial planning tools that can hopefully slow down the depopulation process and, at best, represent a restart for a large part of the territories currently in crisis. To do this, the NSIA asked the municipalities to cooperate by sharing limits and opportunities in a strategic sense, reviewing the territorial governance from below and making it more appropriate to the territory. The suggestion is not to think of a territory limited by administrative borders, but rather of a territory that "talks" given the characteristics of homogeneity present in the neighboring municipalities, and to address the capacity—even at an institutional level—for dialogue and participation, in order to activate the mobilization of the population and resources both internally and externally [32].

The cooperation process thus initiated a geopolitical process on a local scale by activating phases of dialogue between municipalities which, over a couple of years, led to the creation of functional aggregations supported by specific area strategies. The latter had within themselves all of the useful elements through which it would have been possible to understand how the municipalities belonging to the same aggregation would have acted strategically to counteract the phenomena of socioeconomic and environmental "retreat". In this way, it was possible to witness the creation of the project areas (PA,)—each of which, by proposing its candidacy to the various regional commissions, described its strengths and what strategies it would have implemented to reactivate the socioeconomic fabric. Having collected the applications of all of the Italian PAs, the Technical Committee for Inner Areas (TCIA) selected those areas that presented more difficulties than others in guaranteeing the citizenship rights of their residents, as well as presenting high territorial problems—such as hydrogeological instability and loss of utilized agricultural area (UAA)—and problems of a demographic nature, such as depopulation and aging; nevertheless, contexts rich in natural and cultural resources were found [33], in which history and

environment were potentially capable of supporting the collective memory and identity of places, allowing the design and construction of a development that was local in resources, global in relationships, and self-sustainable in modalities [34]—all fundamental elements for a healthy socioeconomic restart, and for more advanced tourism. The TCIA has thus selected, to date, as many as 72 national project areas, 5 of which are in Sicily, for a total of 65 municipalities out of 391, including urban poles, but leaving 80% of the island out of institutional improvement interventions, including the area of our interest. All of the area strategies put in place by the PAs—useful among other things for the implementation of the Framework Program Agreements (APQs), which consist of the necessary tools for the implementation of inter-institutional cooperation—have common elements, including, for example, a strong characterization linked to the agricultural component, supported by the fact that the municipalities of the inner areas are characterized by a strong specialization in the primary sector, ennobled by excellent production [27,35] and, thus comprising territorial capital enriched also by considerable environmental, naturalistic, landscape, and cultural heritage. On the one hand, these elements can be considered strategic for the self-sustainability of the improvement and strengthening of the local system, while on the other hand, they can be functional to the creation of a unified tourism system. It should be noted that these considerations can also be extended to those territories that are not beneficiaries of the NSIA interventions, starting from the assumption that territorial continuity is evident from the point of view of territorial characteristics and attitudes, even if it varies in its presence; this is the case for our area of interest which, although presenting the characteristics of the inner areas, does not have project areas within it, possessing many of the elements mentioned for the creation of a unitary tourist offer. In fact, starting from the coastal areas that, from the international tourist point of view, are already consolidated realities—such as the cluster of the municipalities of Giardini Naxos, Taormina, and Castelmola—it is strategically possible to connect the neighboring coastal municipalities and the innermost municipalities, providing for territorial mending interventions, such as hiking trails and slow mobility systems.

### 2.4. The Case Study: The Taormina–Etna District (Distretto Taormina Etna)

In Sicily, long before the NSIA and its application, there were two other important phases of territorial intervention policies: the Territorial Pacts, and the Tourist Districts. The Territorial Pacts consisted of the implementation of a program of interventions reserved for depressed areas (L. 104/1995), while the Tourist Districts can be identified as socio-territorial entities characterized by a historically and geographically determined territory (L.R. 10/2005). These two phases of territorial intervention are important because, thanks to these tools, it is possible to understand the birth and evolution of the superstructures of economic–political intervention at the basis of what would become known as the Taormina–Etna District (DTE) (see Figure 1).

Currently, all of the public bodies present in the territory of the 60 municipalities falling within the area are part of the DTE; these are distributed among the metropolitan cities of Catania and Messina: the Universities of Catania, Messina, and Enna, the regional parks of Alcantara, Etna, and Nebrodi, the Chambers of Commerce of Catania and Messina, the Catania Airport Company, and the Circumetnea Railway for the public part, and almost 300 companies, banks, employers' organizations, and associations for the private part. This rich group has found fruitful collaboration in, among other things, supporting and promoting the main sectors of the area—namely, tourism and agri-food. The current situation, including the coronavirus pandemic, has greatly reduced the operational technical capacity to intervene in support of the operational realities within the district, forcing both the consolidated tourist realities as well as the more marginal municipalities to rethink new forms of enhancement and socioeconomic support that correspond more to the new needs deriving from a radical change in the method of tourist use.

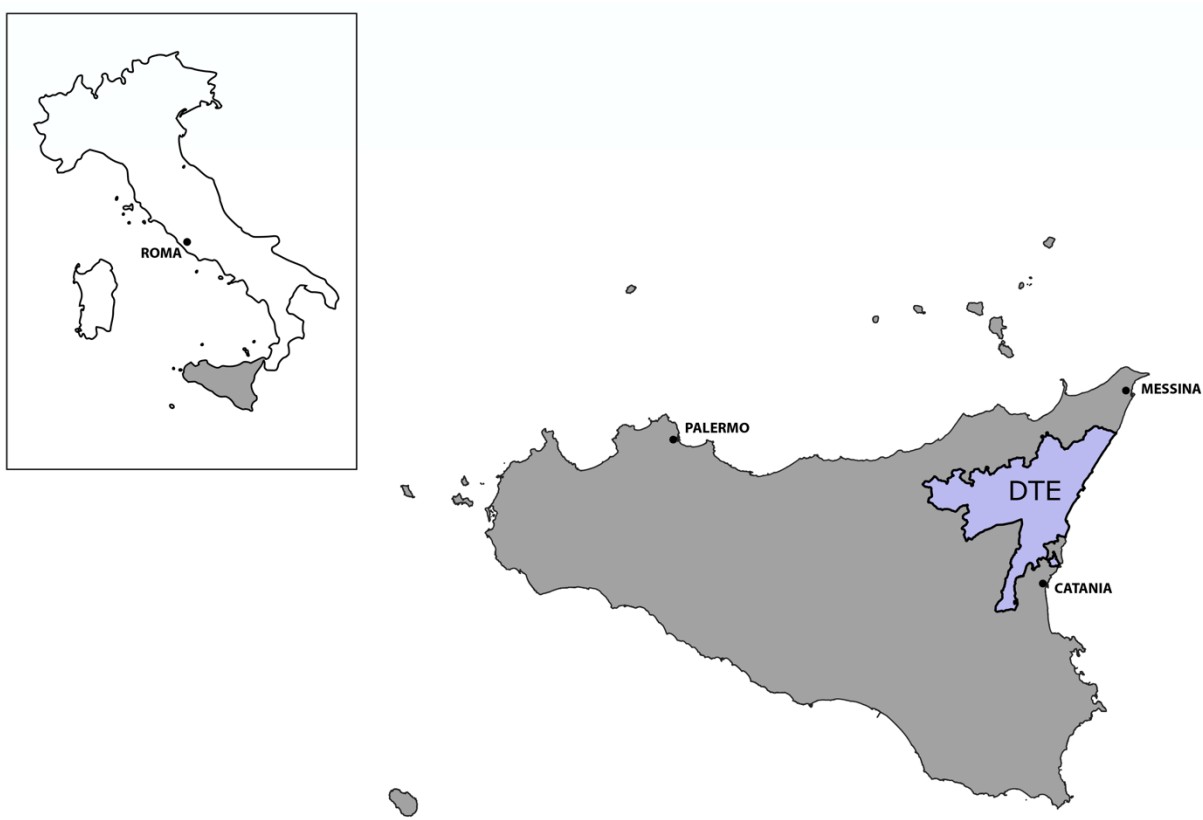

**Figure 1.** Territorial classification of the Taormina–Etna District (Distretto Taormina Etna—DTE).

Therefore, by applying the NSIA paradigm mentioned above to the district, it was possible to identify within the DTE area which municipalities had the characteristics of urban centers or inner areas (see Figure 2a). This mapping has allowed us to observe the structural endowment of the aforementioned municipalities, and to understand, through specific indicators [30], some advantages or limits in participating, in a reticular sense, in hypotheses of territorial mending, such as slow mobility paths. In fact, by applying the criteria of the macrocategories as identified by the Technical Committee for Inner Areas (TCIA), it was possible to identify that out of the 60 municipalities adhering to the DTE, two-thirds possess inner area (IA) characteristics, and the remaining 20 are classifiable as "centers". The latter possess the characteristic of being almost exclusively coastal municipalities, with the exception of the so-called "belt" municipalities of the metropolitan cities to the north (Messina) and south (Catania) of the DTE, however external to them. By applying the more detailed NSIA classification—with reference to the travel time for the use of services by citizens, but also to new and potential tourist flows—we can further articulate the observation by noting how, in addition to the aforementioned belt municipalities (20), there are as many intermediate municipalities—16 peripheral and 4 classifiable as ultra-peripheral—located in the most "internal" part of the district territory, which are also characterized by being entirely classified as mountain municipalities (see Figure 2b).

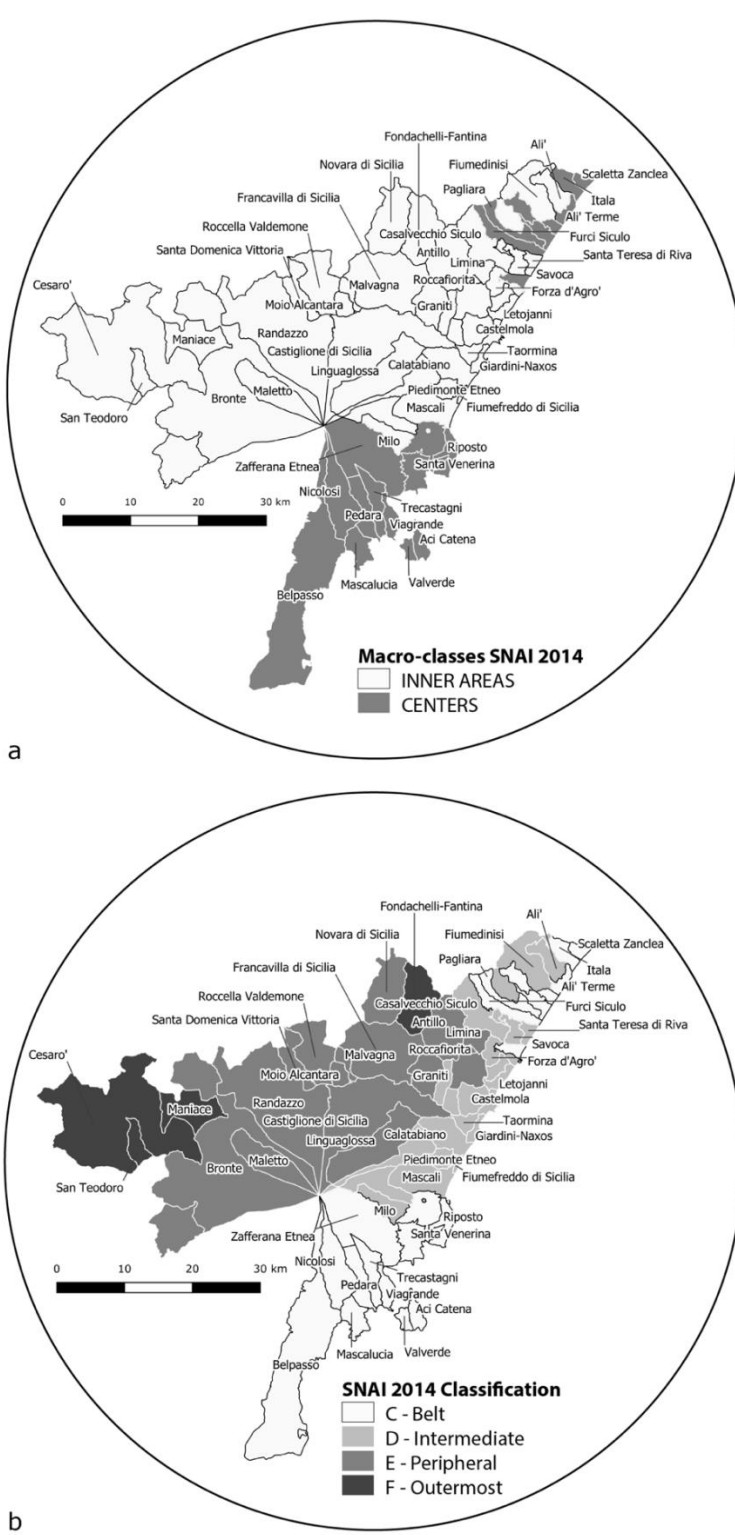

**Figure 2.** (**a**,**b**) Observation of the DTE through the NSIA.

Out of the total of 60 municipalities, 47 are characterized by a population reduction trend, while only 13 show a countertrend; in both cases, there are no significant phenomena, and the variations in the resident population are of a few tens of units. However, it remains alarming how the trend towards depopulation continues to occur in most of the small and very small internal municipalities, in our case almost all falling within the former province of Messina. It should also be emphasized that as many as 14 municipalities are below

the threshold of 1000 inhabitants, and another 25 have a population of between 1000 and 5000 inhabitants, while all of them register a population loss, with the exception of two small municipalities—one coastal and one mountain—both in the sphere of gravitation of the tourist pole Taormina–Giardini Naxos–Castelmola, from which they receive tourists, and to which they provide beds and young people of working age.

Even from the point of view of the aging of the population, alarming data can be recorded in some municipalities of the Messina area, where the proportion of elderly to young people often reaches a ratio of three to one, with cases in which it reaches a ratio of four to one. This ratio highlights the number of elderly people (aged over 65) compared to the number of young people (aged between 0 and 14), where values above 100 indicate a greater presence of elderly subjects than very young ones. Even in the presence of population loss, especially that of working age, the municipalities of the district are home to a significant proportion of the population, with ~350,000 inhabitants, and an average population density of 363 inhabitants per square kilometer, for which the reduction of essential services to citizens also tends to aggravate the condition of production systems. Population density is a useful indicator for determining the impact that anthropogenic pressure exerts on the environment, and is an expression of the degree of crowding in an area. In our case, this indicator is very high, and the reduction of essential services to citizens represents a burden for a large number of inhabitants. The local economy is strongly characterized by agriculture and small manufacturing, with little specialization.

On the coast, on the other hand, tourist activities are almost always concentrated around the larger centers, thus creating socioeconomic polarizations, to the detriment of the innermost areas. This is the case for the municipalities of Giardini Naxos, Taormina, and Castelmola, which together constitute a tourist cluster, in sharp contrast with the rest of the municipalities of the district, possessing a very high tourist density in addition to the homogeneous characteristics of the tourist offer (in fact, all three are classified by the ISTAT as municipalities with maritime, cultural, historical, artistic, and landscape characteristics) [36] (see Figures 3 and 4).

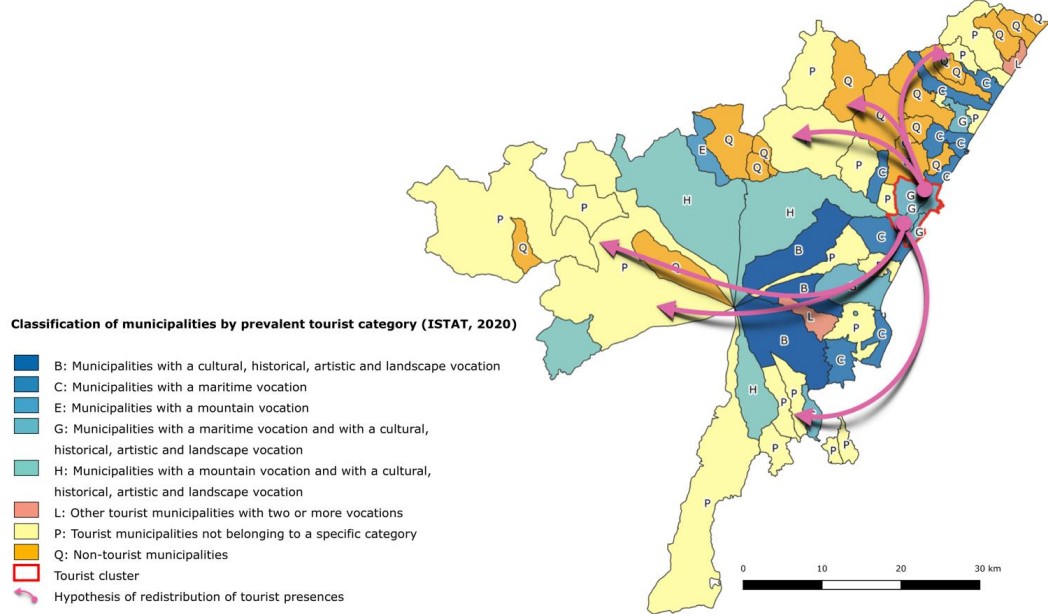

**Figure 3.** Classification of municipalities belonging to the DTE by tourist category.

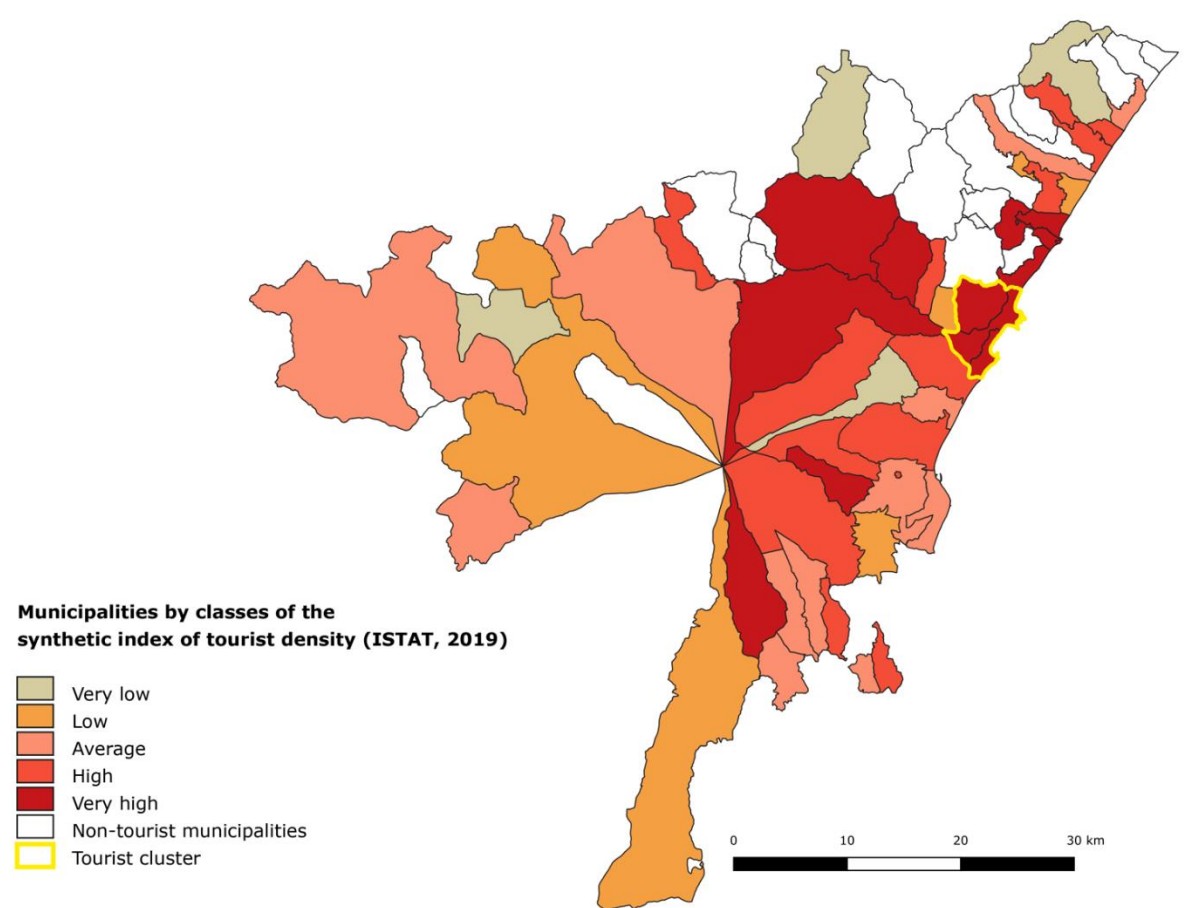

**Figure 4.** Classification of municipalities belonging to the DTE by tourist density classes.

In an attempt to counteract the trend towards depopulation, to rebalance the tourist presence in a wider territory than the identified cluster (see Figure 4), we aim to propose a strategy capable of relaunching the innermost areas, reactivating the territorial capital owned by them or potentially available [37]. Our strategy involves the creation of a slow mobility route from the coast that may lead to the discovery of the internal territory through the metaphor of the "cultural corridor" [38], comprising place-making projects that aim to foster equitable community development through the celebration of local cultural heritage as a stepping stone for further economic investment in a particular geographic area [39]. In our case, this could be represented by the Alcantara River, around which can be identified elements of great attractiveness, as well as the services necessary for mobility and the usability of places, according to the modalities that will be discussed below.

## 3. Results

### 3.1. A Strategic Vision of Integrated Development in the Taormina–Etna District

In order to structure an integrated system of "slow tourism" in the Taormina–Etna District, the interconnection of intermodal infrastructure for light mobility has proven to be an indispensable prerequisite, hence the proposal to implement a territorial co-planning strategy to plan a cyclable/walkable greenway that acts as a cultural corridor to provide multidimensional community actions for the widespread enhancement of territorial capital [40]. What we want to demonstrate in this study is how adopting an "integrated strategy" is the best way to give shape to horizontal governance, building around it a multifaceted strategy capable of acting on the awareness of the patrimonial value of "common goods", sharing "rules of use" on which to test the creativity of the local community [25]. The aim is to prepare a strategic plan structured around two closely related participatory macroprocesses: a "structural–identity" process, entrusted with

the task of reconnecting the ecosystem processes of a community, identifying the resilient co-evolutionary connections on which to focus concerted action [41]; and a "strategic–operational" process, which is entrusted with the task of implementing the projects within a unitary framework, wherein the catalytic actions in which to engage all of the actors of the local partnership are clearly highlighted.

An integrated strategy capable of supporting the coordinated implementation of cycling itineraries dedicated to "slow biking", capable of guiding the visitor from lowland sites to high-altitude sites in a succession of stages in which the services provided (points of tourist attraction, refreshment, assistance, information, accommodation) are connected and digitally mapped by advanced georeferencing systems [42]. In particular reference to cycling routes, there is no real definition by which to identify them, but they can be distinguished as small journeys that can be divided into stages, characterized by one or more unifying themes, which can be undertaken in a linear manner—sometimes reticular or areal—affecting a more or less vast territory [43]. The main types of cycling itineraries can be divided as follows:

-   Network itineraries: these are structured by defining contiguous routes through the territory according to the different themes identified: socio-cultural, morphological or agroforestry characteristics, architectural elements, etc.;
-   Linear itineraries: these are presented as paths defined by a naturalistic or historical route with the function of a commercial, religious, or military route that maintains roads, toponyms, and architectural structures dating back to the road infrastructure that defined it (pilgrimage and wine-trading roads, caravan routes, the Street of the Mills, etc.);
-   Area itineraries: these are comparable to networks of goods, being structured on routes in stages aggregated by a theme that does not present relations of territorial continuity;

Precisely by virtue of the fact that a process of territorial regeneration is by its nature linked to the shared recognition of critical issues and opportunities, the participatory territorial reading constitutes an important motivation for the choice of the "point of attack" on which to concentrate the design strategy, considering:

-   The holistic character: that is, while evaluating specific aspects, the planning must have transversal effects on the various components of the territorial fabric (socioeconomic, environmental, institutional, etc.);
-   The multiscalar character: that is, the planning action in determining the strengthening of the local dimension, which will help to make the strategy flexible and capable of differentiating services on a larger scale (international), starting from the urban centers of reference;
-   The nature of environmental sustainability: in particular, the interventions proposed to maximize the benefits to the communities must guarantee the conservative maintenance and safety of the already existing connection infrastructure.

Ultimately, it is a question of identifying two sets of objectives with respect to the construction of the design strategy: the first is of a general nature (strategic objectives), aimed at demonstrating the way in which the project adheres to the emergent problems in the reference system/context, with specific reference to impact in terms of medium- and long-term benefits; while the second (specific objectives) is aimed at offering a direct benefit to the recipients, based on the services that the project offers in terms of products. Regardless of the type of action, in terms of design, the identification of project needs and objectives is the main priority, from which all of the subsequent processes of elaboration of the integrated development strategy prepared in this work are derived.

### 3.2. Analysis of the Critical and Potential Elements of the Taormina–Etna District

The recursive practices of human action in the region leave traces of different ways of residing in and organizing space, creating a complex schedule of structures inscribed in a network of meanings that remain resilient despite the progressive dismantling of

inherited landscapes [44]—a hermeneutic archive of connective networks that, if investigated, reveals the social strategies and identities that have occurred in the various historical phases, underlying the existing behavior of local communities [45]. The recontextualization of the co-evolutionary interrelationships of territorialization processes, whose rules of use are contained in the long-lasting relationships between contextual knowledge and artefacts, becomes functional to the acquisition of the original "identity matrix", useful for understanding the configuration of today's landscape and projecting it towards a sustainable future [46]. In a process of regional enhancement centered on the interests of a specific community, the reconversion of the individual regional components can only be proposed after having identified the multiple historical connections with other elements—a dispersed plot of physical and natural factors that can be found in our overwhelmingly man-made landscapes, which we perceive to be significant, albeit apparently secondary. The most suitable choice of direction for cognitive investigations towards a regional project of significant reconnection with the original regional character from the point of view of the community—according to the prescriptions of the European Landscape Convention of 2000 (regarding the objectives of landscape quality based on the perception of the community) and the Faro Framework Convention of 2005 (on the value attributed to cultural heritage by society)—seems to be to link all of the phases of specialist study to participatory geo-mediation actions, in order to define shared strategic guidelines [47].

The pre-feasibility study at a methodological level starts from the analysis of the context according to a multimethod approach that is based both on careful research of the studies carried out and on direct observation, with in-depth analysis aimed at evaluating the perceptions of stakeholders through a qualitative perspective, based on 50 semi-structured interviews with institutional actors, entrepreneurs, citizens, and representatives of local communities. For the construction and maintenance of short networks—intended as a network of actors within the region—in terms of "who does what" and "where" it is done, the project, in a broader perspective, sees the possibility of collaborating with external actors, who give rise to long networks (Table 1).

The results that emerge from the analysis of the context represent a region characterized by strong peripheral conditions, which manifest themselves in functional deficits linked to the loss of importance of traditional economic activities, environmental deficits that highlight the increase in hydrogeological instability along with the growing difficulty of ensuring the long-term resilience of the agro-pastoral fabric, and relational deficits evident in the loss of traditional forms of socio-cultural aggregation, in the face of which the reactivation of community resources appears to be fundamental to any prospect of change (Table 2).

The new fields of territoriality are based on the self-recognition of the "territorial system" between the sphere of environmental/cultural assets and the variegated sphere of the inhabitants/users, through tools of participation tending to the self-governance of the patrimonial common goods. The intervention as a whole proposes an innovative model of self-sustainable local development, and is based on crucial concepts such as integration, connection, culture, attention to socially fragile categories, innovation, and attention to the environment.

Faced with the critical issues mentioned above, it is possible to identify a series of exclusive factors that, together with a rich cultural, archaeological, and natural heritage, constitute a potential foundation on which to build integrated strategies for tourist offers linked to experiential receptivity and especially attentive to local food and wine production, able to stimulate an "active welcome" based on the "knowhow" of the local community (Table 3).

**Table 1.** Project stakeholders.

| Scope | Category | Main Actors |
|---|---|---|
| Local | Local institutions | - Common<br>- Local action group (GAL)<br>- Tourist district<br>- Museums<br>- Destination management organization (DMO)<br>- River Park Authority<br>- Ecclesiastical institution |
| | Associations | - Cciaa<br>- Trade associations<br>- Onlus<br>- Cooperative<br>- Foundations<br>- Committees<br>- Voluntary organizations<br>- Cultural venues<br>- Tourism promotion associations |
| | Training institutions | - Schools |
| | Manufacturers | - Entrepreneurs<br>- Craftsmen |
| | Private | - Experts, connoisseurs<br>- Consultants, technicians<br>- Inhabitants<br>- Travel agencies<br>- Tour operator |
| External | Public institutions | - Regional departments<br>- Ministry |
| | Training institutions | - University |
| | Entrepreneurs | - Companies<br>- Associations |
| | Private | - Consultants, experts<br>- Cultural operators<br>- Travel agencies<br>- Tour operator |

Source: constructed by the authors.

**Table 2.** Critical elements of the Taormina–Etna District.

| Critical Elements | | |
|---|---|---|
| **Region** | **Society** | **Economy** |
| • High hydrogeological risk linked to the morphology of the region;<br>• Significant seismic risk linked to the lithology of the soils;<br>• The infrastructure for mobility between the coast and the internal area is obsolete; moreover, the few cycle paths have not been created in a coordinated and uniform manner;<br>• State of deterioration of the pre-existing medieval and rural architectural structures due to the abandonment of the historical building heritage and the absence of recovery interventions;<br>• Loss of the identity of the locations due to interventions left to the whims of individuals and professionals, without a cohesive plan. | • Depopulation;<br>• High old-age index, with peaks exceeding 200%;<br>• Negative demographic trend (> emigration) due to the lack of services and study and work opportunities;<br>• Decrease in the foreign population due to a lack of services and difficulty in integration;<br>• Increase in unemployment, with particular regard to the youth population, due to the lack of job opportunities in line with the demands of young people and the crisis in the production sector;<br>• High rate of early dropout from the school system due to inadequate training provisions and difficulty in traveling;<br>• Insufficient health and social welfare facilities. | • Crisis in the agricultural, manufacturing, and construction sectors; transition to the service sector;<br>• Crisis in the commercial sector, which has recorded a contraction of neighborhood shops in the face of an increase in large-scale distribution outside the area;<br>• Decrease in the spending power of the local population due to the economic situation;<br>• Declining tourist flows (despite an increase in the number of businesses in the tourism sector) due to the lack of an integrated tourism offer;<br>• Micro-sized entrepreneurial realities, which struggle to network with one another. |

Source: constructed by the authors.

**Table 3.** Attributes of potential of the Taormina–Etna District.

| Attributes of Potential | | |
|---|---|---|
| **Region** | **Society** | **Economy** |
| • Ability to develop systemic relationships with the city of Taormina, Naxos, and the internal region implementing connections from the point of view of cycling tourism, focusing on intermodality and the completion and enhancement of hiking itineraries;<br>• High degree of naturalness of the area, with respect to which attractive offers can be developed in the different seasons of the year;<br>• Presence of important water resources (Alcantara) and a vast natural heritage, currently unused;<br>• Presence of niche and high-quality production linked to the dairy supply chain;<br>• Presence of important archaeological and architectural identities, linked to the history of the district. | • Possibility of enhancing and positively exploiting the strong attachment to the region present in the population, in order to stimulate an entrepreneurial spirit that leads residents—especially young people—to undertake economic activities in the region;<br>• Presence of a living social fabric, with important resources in the world of volunteering and the third sector;<br>• Presence of small local communities where relationships and a sense of belonging are still present. | • Possibility of enhancing the peculiarities of the region (typical products, traditions, and culture) in order to develop a rural and experiential tourism aimed primarily at the inhabitants of urban centers;<br>• Possibility of developing cycling tourism thanks to the coordinated implementation of cycling routes, with the offer of accommodation facilities, in order to offer specific services for cyclists (both amateur and expert);<br>• Possibility of intercepting the new internal and external employment demand, in order to develop an offer that is attractive both for young people in the area and for young people/adults who come from urban centers (conversion of farms into agritourism destinations, educational farms, excursion services, etc.). |

Source: constructed by the authors.

## 4. Discussion

*Hypothesis of the First "Tourist Cycling Cultural Corridor" in the Taormina–Etna District*

In the logic of "reactivation" of the district, one of the main potential elements on which to act is constituted by the possibility of reorganizing the relations between the strong attractions of the tourist cluster including the municipalities of Taormina, Giardini Naxos, and Castelmola, and the internal areas, supporting a renewed development model that enhances the identity and distinctive elements of the district, delineating a "bridge" (material and nonmaterial) connecting the mountain and coastal areas (Figure 5).

The project to mend the territory begins at the coast, and heads towards the mountainous areas of the Nebrodi, where one can perceive a central axis consisting of a narrow valley groove created by the erosion of the rocks from the Alcantara River, involving the perimeter of the homonymous Fluvial Park (~11,000 hectares between the Mojo Alcantara valley and the mouth of the river near Giardini), characterized by numerous geomorphological, vegetation, rural, and historical–artistic (towers and castles, Byzantine cube, ancient mills, the Regia Trazzera, etc.) elements that have conditioned its socioeconomic evolution [48]. The Alcantara River, which in its route divides the provinces of Catania (Calatabiano, Castiglione di Sicilia, Randazzo) and Messina (Francavilla di Sicilia, Floresta, Gaggi, Giardini Naxos, Graniti, Malvagna, Mojo Alcantara, Motta Camastra, Roccella Valdemone, Santa Domenica Vittoria, Taormina) is characterized in its initial section by a fiumara morphology, with a wide riverbed occupied by alluvial deposits; from the territory of Mojo Alcantara to near the mouth of the river it is encased for half of its path in basaltic rocks of volcanic origin. In the final stretch, an imposing series of lava flows modeled by the erosive process of the waters create two important geosites: the "Forre dell'Alcantara"—where the waters in correspondence with morphological differences in height create extraordinary rapids near the Cuba of Santa Domenica in Castiglione, Sicily—and the "Gole dell'Alcantara" of the deep gorges with overhanging waterfalls in Motta Camastra [49] (Figure 6).

Following the example of many international realities, the aim is to redevelop the hydrographic segment by structuring cycling paths on existing roads (secondary roads, disused railways, lanes, etc.), and a greenway structured as a "cultural corridor" that reconnects the territorial heritage to the coastal communities in an attempt to stimulate self-sustainable local development actions [38,50–53]. By arranging an integrated offer along the itinerary in stages that involve local operators determined to specialize in cycling tourism by offering specific services and precise information to those who want to discover a portion of the Taormina–Etna District, the intention is to seasonally adjust the flow of visitors.

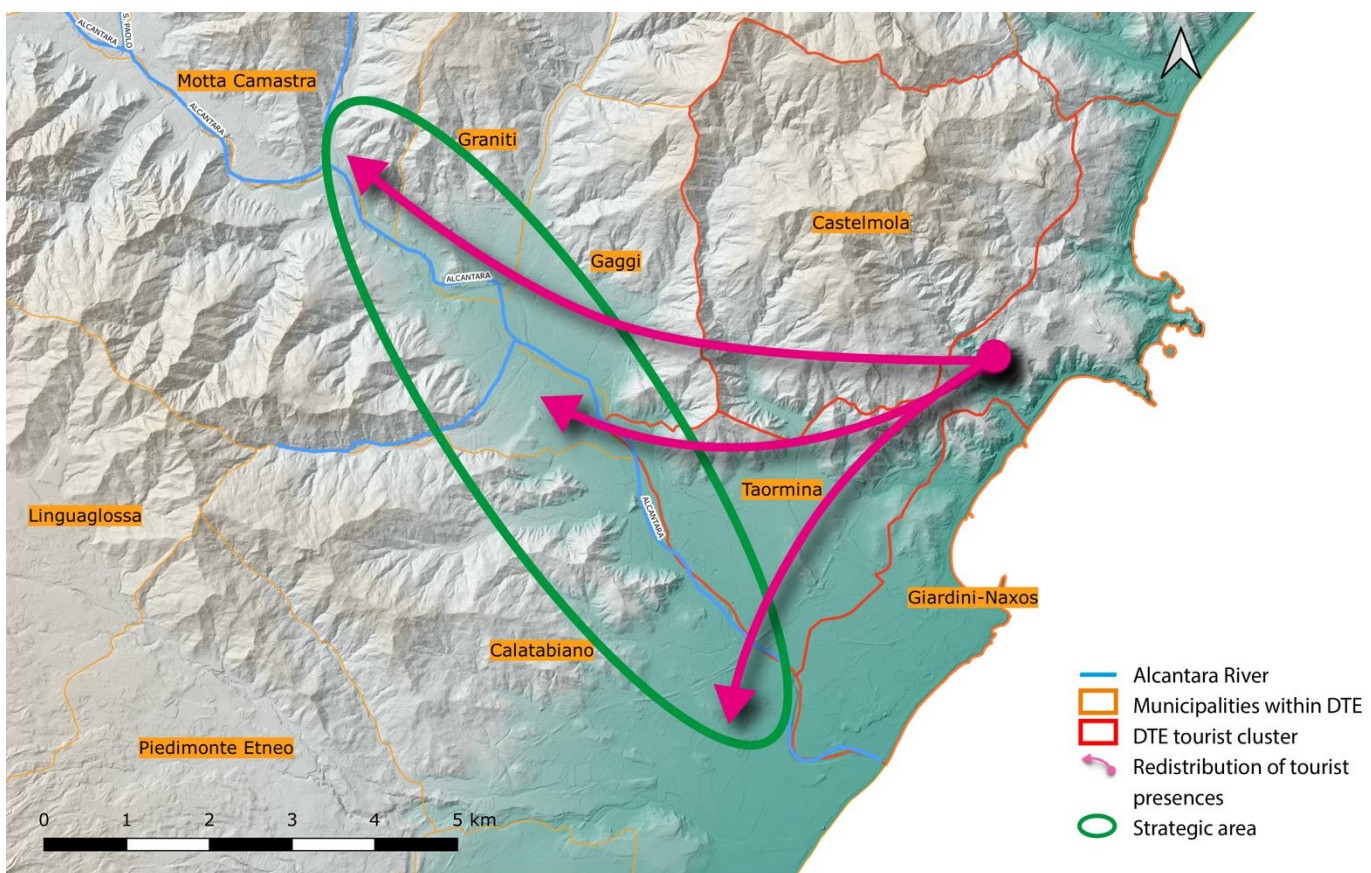

**Figure 5.** Planning of the cycling route.

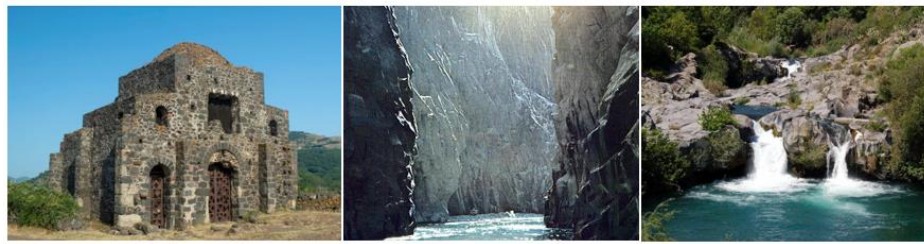

**Figure 6.** Left to right: Cuba of Santa Domenica; Gole dell'Alcantara; Forre dell'Alcantara.

The implementation of such a strategy will be the expression of a public–private partnership that sees the Taormina–Etna District as the leading entity—the body that supervised the activity carried out for the preparation of this study, and which will continue to play the coordinating role during implementation. Consistent with the provisions of the guidelines issued by the Italian legislation for the preparation of the "Urban Sustainable Mobility Plans" (PUMS) of D.M. 397/2017 for urban and peri-urban areas, a strategy has

been defined that provides for the participation of stakeholders in all phases of the design of the cycling route [43]:

- Presentation phase: Contact with the project group, who in the plenary workshop will explain the aims of the initiative with the help of illustrative panels, and will share the articulation of the participatory path;
- In-depth phase: Construction of a SWOT analysis of the values and criticalities of the area affected by the design hypothesis of the cycling route, based on reports from the community involved: participatory inspections, interviews, administration of questionnaires, and interaction via the web (online forum). From the evidence of the strengths/weaknesses and the opportunity/threat analysis (SWOT approach), a vision will be shared among the stakeholders involved, aimed at defining a systemic decision-making strategy [54];
- Proposal phase: In this step, in order to ensure the maximum connection between the partners, the working groups will be organized based on themes identified among the various intervention proposals developed for the design of the cycling path in terms of the services offered, ancillary infrastructures, enhancement of places and/or areas, and technical choices of materials and furnishings. Then, there will be a period of sharing and verifying the work done, where the results of the participatory process are assembled and the final formulation of the cycling path project is produced, which could be formally approved by the municipalities concerned as a preliminary feasibility study for the real design process.

## 5. Conclusions

All stakeholders from the public sector and the travel agencies are very interested in the development of cycling tourism in the analyzed area, emphasizing the coordination of work and the good cooperation that should enable them to achieve their goals. The socioeconomic emergency situation induced by COVID-19 redefines the accessibility of internal areas by strategies as well as the investments in the recent recovery plan [55]. The funds should be addressed primarily to the reconversion of existing infrastructure (disused railway lines, river banks, secondary roads, etc.) by sharing intermodal solutions for soft mobility (trains, bicycles, walking) to envisage an integrated use of internal areas by reducing the social inequalities caused by territorial confinement [56].

Cycling tourism could represent a valid socioeconomic revitalization strategy in areas affected by the coronavirus emergency; cycling develops small-scale economies in the areas crossed by encouraging the seasonal adjustment of the tourist offer. Adaptable to proximity tourism (staycation, holidays near home, etc.), cycling tourism expresses the distinctive characteristics of the low-touch economy—safety, health, distance, and short range.

To enable such development, local actors need to identify this opportunity and collaborate in a vision of collective interest defined through participation and involvement in the specific territorial context [18]. Furthermore, it is necessary to consider the obligatory nature of the involvement of citizens in the process of drafting the "Urban Sustainable Mobility Plans" (PUMS) for urban and peri-urban areas, as established by the guidelines issued by Italian legislation (Ministerial Decree 397/2017) in compliance with European legislation. The PUMS focuses on people, and the satisfaction of their mobility needs, following a transparent and participatory approach that provides for the active involvement of citizens and other stakeholders from the very beginning of the process. The subjects involved in the participatory planning of the cycling route—both local actors and local institutions—must find suitable grounds for the exchange of information and for the sharing of proposals, building a common vision of the actions to be carried out by combining entrepreneurial innovation (e.g., cycling tourism) and sustainable development [44], favoring both the coordination between actors, subjects, and decisions, and the growth of cooperation, thus constituting a program of actions to be implemented, as well as a reference vision in which local actors can recognize themselves, and which can lead to real innovations in rural areas that aim to combine tourism diversification and sustainable development in order to



create synergies between tourism diversification (e.g., cycling tourism), entrepreneurial innovation, and sustainable development [57].

Furthermore, activities related to cycling tourism can become an important driving force for the development of inland areas, far from the well-known tourism destinations, where an essential factor for success is the ability to structure an integrated tourism system to create and strengthen cycling networks. This requires a more sustainable and participatory approach to tourism planning, decision making, and management, which remains one of the key challenges faced by destination planners at all levels. We must imagine a mobility model that allows the combination of different means of active mobility (e.g., train and bicycle) to develop a winning tourist offer in areas with characteristics such as those of internal areas that extend beyond administrative and territorial boundaries.

Our research is not without limitations. Firstly, the qualitative approach based on a few interviews with stakeholders and public head offices limits our ability to generalize our findings, which we need to be cautious in applying to other contexts. A quantitative study could help to overcome limitations of generalizability, and be used to assess a correct engagement of stakeholders to determine their influence in the implementation process.

We hope that our findings will encourage further research on practices of revitalization of internal areas that operate internationally, to determine whether differences in strategic processes influence implementation in these areas. Our research also contributes to practice and policy. In fact, our findings may help policymakers to better understand the roles of different stakeholders in green practices such as cycling tourism.

**Author Contributions:** Conceptualization and methodology, G.P., D.P., and G.R.; software GIS, G.P.; data curation and writing—original draft preparation, G.P., D.P., and G.R.; writing—review and editing, G.P., D.P., and G.R.; funding acquisition, G.P. and D.P. All authors have read and agreed to the published version of the manuscript.

**Funding:** This research was partially funded by Research Programme: Unict 2020–2022 PIACERI-Linea 2, and OPEN ACCESS Linea 4.

**Institutional Review Board Statement:** Not applicable.

**Informed Consent Statement:** Not applicable.

**Data Availability Statement:** Not applicable.

**Acknowledgments:** The authors acknowledge Oreste Lo Basso, DTE senior project manager, for his support and the materials provided for the research.

**Conflicts of Interest:** The authors declare no conflict of interest.

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
