# Peer review of "Cycling Tourism and Revitalization in the Sicilian Hinterland: A Case Study in the Taormina–Etna District"

_sustainability, doi:10.3390/su131810022_

Round 1

Reviewer 1 Report

This is an interesting paper that is way too long for a very long and descriptive review of what might happen in NE Sicily. Perhaps because it is not an empirical piece, it runs long on editorializing. 

The paper in its current form fails to connect concisely with larger kinds of literature: sustainable tourism, cycling tourism, slow tourism, cultural-heritage tourism, among others.

First and foremost, the long Italian sentences are beleaguered by tortuous passive voice construction in this English version. Use your Spelling-Grammar check to get 1) your passive voice sentences down to less than 10 percent 9<10%), and 2) your sentence length averages under 20 words per sentence. The sentences are so long and so tortured with multiple clauses, that --as a native speaker-- by the time I got to the end of many run-on sentences, I had to stop and re-read the sentence to regain the meaning. 

I am sympathetic to professional writing in a language other than one's own. However, a technical style of writing is required here, and the more Baroque Romance Language style of long, passive-voice construction has to be changed. This will allow the authors to distill their argument into more concise language. 

Remember: you are very close to this case study. One senses that you feel compelled to us absolutely everything about the behind-the-scenes deliberations behind this cycling/tourism project. You can easily edit out 50% of the word count and distill your argument. Please try to do this.

Here are some more specific comments that must included in any revision.

  1. The title is too long and verbose. How about "Cycling Tourism and Revitalization in the Sicilian Hinterland"? 
  2. In an economic geographic sense, you are talking about the hinterland. The use of territory is too much of literal translation from the Italian. The word 'territory' is also used in places where "region" would be more precise. I'll point out a few specific lines below.
  3. Line 9 "strategy as the strategic..." ?????? What? Rewrite and shorten. Tautologies here and elsewhere. 
  4. lines 22-24. Very wordy. Re-write.
  5. Line 45. Awkard. so-called "slow". It is not so-called "slow." It is slow tourism just like the slow-food movement (which originated in Italia and to which you make no reference, I believe0. 
  6. Lines 41-50. This is a ridiculously long sentence that could be broken up into multiple sentences or else greatly condensed. Indeed, the paragraph is very long and consists of just two sentences. Rewrite and shorten here and elsewhere. 

Line 74. I think 'hinterland' is better here. It reduces all the verbiage too. See this definition of hinterland from Merriam-Webster's dictionary; 

hinterland

 Log In  hin·​ter·​land | \ ˈhin-tər-ˌland  -lənd \

Definition of hinterland

1a region lying inland from a coast 2aa region remote from urban areas ba region lying beyond major metropolitan or cultural centers  
  1. line 117. This is huge! If Italy has 7900 municipalities (counties?), consider that the US which is many times larger, has about 3000 counties. This statistic, alone, sounds like a planner's nightmare. The sure volume of interests is overwhelming. There are lots of planning implications here about the nightmare of trying to generalize from so many data points.
  2. line 165. The reference [14-7] is unclear. Is it page 7 in reference 14 to which you refer? Is this the journal's citation format? I defer to the editor on this. Or, is it a span of references from 14 through 7?
  3. Figure 1. Good map. Add a scale. Add major cities: Palermo, Etna, Siracusa. 
  4. Figure 2a and 2b. Lovely maps but too hard to read. Increase the scale and set them apart as separate (larger font) images.
  5. Lines 234-42 are good examples of whether all this background information is necessary. Again, detach yourself more from all this detail and aim for the big picture. Your readers are not from Sicily and they want to know the big-picture takeaways from this proposal to use cycling to revitalize a poor area that has been hit by Covid. 
  6. Figure 3 is wonderful! Question: Is 'hypothesis' the correct word here? Is it not a proposed route?
  7. The discussion of 'cycling' does not happen until the middle of the paper. This should come sooner. How can your case study inform other cycling and tourism routes and planning? That is the big picture. When the reader is done with your paper, s/he wants to know what lessons does the Sicilian case study offer my region. The authors need to keep this in mind; they don't want to be experts on the Italian example in this paper. If the authors keep this in mind, they can easily reduce the 'background' information, and then consider the generalizability to other places. 
  8. Line 356. 30% increase from when to when? Unclear.
  9. Line 370. Here and elsewhere: Search for the word 'administrations" because, again, it may be a literal translation from Italian, but the word "jurisdictions" (e.g, separate government entities that exercise legal control in some manner over a well-defined political space) might be more appropriate.
  10. lines 458-462. Another example of a sequence that is too long. What does it mean? Is it essential?
  11. Line 489. Here and elsewhere, the word "territory" is used in many places when the human geographic construct is really "region.'
  12. Line 486. The word/concept of 'stakeholders' is important. here and elsewhere you refer to internal/external stakeholders. Fine, but where do you define them? Perhaps a short table listing in two columns, who these stakeholders are? We can deduce who they might be. You use the word 'stakeholders' five times in the paper, but hey are never really identified. 
  13. Line 496. Elaboration by...? Incomplete
  14. Table 1. Do you mean territory or region? I think the latter. 
  15. Table 2. These are "attributes' which could be reworked into the title which would greatly reduce the length of the table title.
  16. Line 504. Again, incomplete attribution "Elaborated by..."
  17. Line 512 immaterial should be nonmaterial
  18. Line 521. How is geomorphological different than natural? By natural to you mean 'ground cover' or 'vegetation'? Specify the components of "natural" which is quite broad.
  19. Lines 569-582. Another extremely long sentence. 
  20. Ways to connect the case study to broader, more generalizable processes are seen in your reference 20 about recreational and cycling routes. Build on these non-Italian processes to contact to a broader set of literatures.

Summing up, you need to edit out 50 percent of the text here. Shorten sentences. Connect to bigger-picture ideas.

Lastly, although the paper is structured as an empirical case study, the paper really is normative about what could be, what might be. You are asking the reader to read a lot about a proposal that has yet to take hold, are you not? 

Author Response

We are grateful for your valuable comments, thank you!

Reviewer 2 Report

Please add limitations of the research and comments about future research

Author Response

(The authors gave the same response as above.)

Reviewer 3 Report

In the introduction, more elaboration is needed about the impact of Covid 19 on tourism;

More literature review is needed to put the paper in an international context;

More description is needed on the tourism potentialities of the selected Taormina Etna District, we need also pictures of the district; population, economic activities, the urban pattern and so on;

Figure 1 should be changed with another one which shows other cities especially the capital city of Rome;

In P.6, can you state the reasons of depopulation in the district selected;

The strategy of slow tourism or cultural corridor based on cycling, can you reinforce this solution by some international or local case studies;

Can you replace Figure 05 by a much cleared map that shows the planning of the cycle route;  support this by pictures;

Please support your paper by more pictures on the city and the district, to show the main cycling route and the different stops if any;

We cannot talk about tourism without pictures;

Author Response

(The authors gave the same response as above.)

Round 2

Reviewer 1 Report

develops a development strategy; how about "proposes a development strategy" to avoid using derivatives of develop in the same sentence?

These are not proper nouns, so lower case please: "Public Sector and the Travel Agencies "

Congratulations on a nice revision! 

All maps require a scale (especially since Euclidean distance is important to bikers!)

Author Response

Thank you again for all the suggestions.

We followed them in the new revision.